# MOF-808 as a Highly Active Catalyst for the Diastereoselective Reduction of Substituted Cyclohexanones

**DOI:** 10.3390/molecules27196315

**Published:** 2022-09-25

**Authors:** Hans Hilmar Mautschke, Francesc X. Llabrés i Xamena

**Affiliations:** Instituto de Tecnología Química UPV-CSIC, Universitat Politècnica de València, Consejo Superior de Investigaciones Científicas, Avda. de los Naranjos s/n, 46022 Valencia, Spain

**Keywords:** Zr metal–organic framework, Meerwein–Ponndorf–Verley reaction, heterogeneous catalysis, substituted cyclohexanone

## Abstract

Zr-containing MOF-808 is an excellent heterogeneous catalyst for the diastereoselective Meerwein–Ponndorf–Verley reduction of substituted cyclohexanones. The presence of substituents at the 2 or 3 position of the cyclohexanone ring strongly drives the reaction towards the formation of one of the two possible isomers. For 3-methyl cyclohexanone, the available space inside the MOF pores allows the formation of the bulkier transition state leading to the thermodynamically stable 3-*cis*-cyclohexanol. For 2-methyl cyclohexanone, the reaction rate is much slower and the final diastereoselectivity depends on the size of the alcohol used. Finally, reduction of 2-phenyl cyclohexanone is considerable faster over MOF-808 than for any other catalyst reported so far. The large size of the phenyl favors the selective formation (up to 94% selectivity) of the *cis*-alcohol, which goes through a less hindered transition state.

## 1. Introduction

The Meerwein–Ponndorf–Verley reaction (MPV) [1,2,3] consists of the reduction of a carbonyl compound (an aldehyde or a ketone) to the corresponding (primary or secondary) alcohol using a sacrificial alcohol as a source of hydride ions, as shown in Figure 1.

First reports on MPV reaction used aluminum alkoxides as catalysts and the reaction was described to be reversible. The opposite reaction, i.e., oxidation of an alcohol to a carbonyl in the presence of a sacrificial aldehyde or ketone, is known as Oppenauer oxidation. The MPV reaction is chemoselective, so that carbonyl groups in a molecule can be readily reduced in the presence of other easily reducible functional groups, such as unsaturated C-C bonds, esters, and nitro groups, which are not altered during typical MPV reaction conditions. This is a clear advantage of MPV versus other reducing agents, such as NaBH_4_. In oxidation reactions, overoxidation of the carbonyl compounds to carboxylic acid does not take place.

However, a clear drawback of MPV is that it usually requires large (even stoichiometric) quantities of aluminum alkoxides. Therefore, development of new efficient catalysts for MPV reactions is an area of active research. Novel catalysts have been explored for this reaction based on transition metal complexes of titanium, zirconium, hafnium, or iridium; lanthanide alkoxides and several heterogeneous systems. These include various metal oxides (γ-Al_2_O_3_, [4] La_2_O_3_, [5] MgO [6], and hydrous ZrO_2_ [7]), zeolites and mesoporous silicates containing Al, [8] Ti, [9] Sn, [10] Zr, [11] Nb or Ta [12] centers; as well as metal–organic frameworks (MOFs). In particular, various Zr- and Hf-containing MOFs have been successfully used as catalysts for the MPV reduction (and Oppenauer oxidation) of carbonyl compounds [13,14,15,16,17,18,19,20,21].

Given the pro-chirality of the carbonyl group, the MPV reaction can be used in diastereoselective reductions. Interestingly, diastereoselectivity can be achieved in many cases even without the need for chiral reducing agents or catalysts, due to chelation or steric reaction control. For instance, non-chiral Al(O^i^Pr)_3_ have been used in the diastereoselective synthesis of *anti*-diols [22] or chiral ephedrine analogs, [23] while we have described the use of achiral Zr-MOF-808 for the diastereoselective synthesis of a number of hydroxysteroids [16,17]. In this sense, substituted cyclohexanones have been extensively used as substrates to evaluate diastereoselective properties of various MPV heterogeneous catalysts, where the observed diastereoselectivity was usually attributed to steric effects. Thus, aluminum [24] and M^4+^ (M = Ti, Sn, Zr) [9,10,11,25] zeolites have been used for the reduction of various substituted cyclohexanones, including 2-methyl, 3-methyl, 4-methyl, and 4-*tert*-buthyl cyclohexanones, affording the thermodynamically less stable alcohol, with the hydroxyl group in axial position. This was attributed to the restricted space available inside the narrow zeolite pores, which drove the reaction preferentially towards the formation of the less bulky transition state. Interestingly, the use of mesoporous silicates (MCM-41, MCM-48, or SBA-15) [11,26,27,28,29,30] with grafted metal-alkoxides yielded the opposite diastereoselectivity; i.e., the thermodynamically more stable alcohol was obtained, due to the unconstrained space inside the much wider pores of those solids.

In view of our previous success in using Zr-containing MOFs for the MPV reaction, herein we describe the catalytic properties of MOF-808 for the reduction of various substituted cyclohexanones, with special emphasis on determining the diastereoselective properties of the material as compared to other catalytic systems previously discussed in the literature.

## 2. Results and Discussion

Zr-containing MOF-808, as well as other catalysts used for comparison purposes, were all prepared as described in the Materials and Methods section, and thoroughly characterized (Appendix A). The general procedure used for the MPV reduction of substituted cyclohexanones is also described in detail in the Materials and Methods section, while specific reaction conditions are indicated in the corresponding Tables and Figures below.

### 2.1. Reduction of 3-methylcyclohexanone (3MeCH = O)

Table 1 summarizes the main catalytic results obtained with MOF-808 and other catalysts for the MPV catalytic transfer hydrogenation of 3MeCH = O, using either ^i^PrOH or 2-BuOH as solvents and hydride sources. With both alcohols, conversion of 3MeCH = O was complete after 6 h under the reaction conditions studied, and 3-methylcyclohexanol (3MeCH-OH) was the only product observed. Therefore, the reaction was fully chemoselective. No conversion of 3MeCH = O was observed in the absence of the catalyst (blank experiment).

Note that when cyclohexanone bears a substituent, two chair conformers can be formed, having the substituent in either the axial or equatorial position. Owing to the higher steric hindrance in the axial position, the substituent is better stabilized in an equatorial position, and this conformer predominates in the equilibrium. Therefore, the most stable isomer of 3MeCH = O is the one having the CH_3_ group in the equatorial position. Thus, two transition states can be considered based on the most generally accepted mechanism for the MPV reduction of ketones over Lewis acid catalysts (Figure 2) [22]. In TS-A, an axial hydride transfer leads to an equatorial -OH group in the final product, while in TS-B, an equatorial hydride transfer leads to an axial -OH group. Thus, each TS leads to the corresponding alcohol, with the -OH group in either *cis* or *trans* position with respect to the CH_3_ group.

As it can be seen in Table 1 (entry 1), when MOF-808 was used as a catalyst, 3MeCH = O was preferentially converted to the *cis*-alcohol, either (1S,3R)- or (1R,3S)-3-methylcyclohexan-1-ol (*cis*-3MeCH-OH). This is thermodynamically the most stable isomer, since the -OH group occupies an equatorial position, while *trans*-3MeCH-OH bears the -OH in the axial position. A very high diastereoselectivity (82%) was obtained with ^i^PrOH towards the *cis* isomer. With 2-BuOH as a reducing agent (entry 2), the *cis* isomer was still the dominant product but the selectivity decreased to 68% at full conversion.

The activity of MOF-808 was then compared with that of other heterogeneous catalysts. Thus, the reduction of 3MeCH = O was very slow over UiO-66 under the same reaction conditions (3% after 24 h of reaction with both ^i^PrOH and 2-BuOH). The conversion sharply increased to 98% after 24 h when the reaction temperature was increased to 120°C, yielding the *cis* alcohol with a 69% diastereoselectivity. The lower catalytic activity of UiO-66 with respect to MOF-808 for the MPV reaction has been previously discussed in detail elsewhere [16,17,18,19]. It is attributed to two main causes: (1) the lower amount of accessible Zr^4+^ sites in UiO-66 (only those associated with lattice defects); and (2) the narrow pore system of UiO-66, which hinders diffusion of reactants and products and the formation of the bulky transition state of the MPV reaction inside the MOF cavities. The differences in catalytic activity between UiO-66 and MOF-808 are more evident for the reduction of 3MeCH = O than for unsubstituted cyclohexanone, due to the larger dimensions of the former compound and to the additional steric hindrance introduced by the methyl group. Table 1 also includes data of the 3MeCH = O reduction taken from the literature (entries 6–8): Zr- and Ti-beta zeolites and mesoporous SBA-15 silica with grafter zirconium propoxide. As in MOF-808 and UiO-66, Zr/SBA produced *cis*-3MeCH-OH as the main product with 75% diastereoselectivity. However, both Zr- and Ti-β zeolites yielded the *trans* isomer with 71% and 70% diastereoselectivity.

The results commented above illustrate well the relevance of confinement and steric effects on the reaction outcome. The space available inside the pores of the solid catalyst determines which one of the two possible transition states fits better, thus driving the reaction to the formation of one alcohol product preferentially. This is a clear example of a transition state selectivity.

According to the results shown in Table 1, two groups of catalysts can be distinguished according to the preferential formation of either *trans*- (Zr- and Ti-β zeolites) or *cis*-3MeCH-OH (MOF-808, UiO-66, and Zr/SBA). In the former group, the limited space available inside the zeolite channels (Ø < 8Å) disfavors the formation of the bulkier TS-A, with an axial hydride transfer. Therefore, the less thermodynamically stable product *trans*-3MeCH-OH is formed, with the -OH group in axial position and in *anti* configuration with respect to the -CH_3_ group in position 3. In the latter group of catalysts, the wider pores (as in mesoporous Zr/SBA) or their pore shape (in the form of cavities instead of channels, as in MOF-808 and UiO-66) allows the formation of the bulkier TS-A, leading to the thermodynamically more stable *cis*-3MeCH-OH.

Therefore, MOF-808 is an interesting catalyst for the MPV reduction of 3MeC = O, since it features a catalytic activity that is similar to that of Zr- and Ti-β zeolites but with the opposite diastereoselectivity, comparable with Zr/SBA. However, UiO-66 is less remarkable than MOF-808, due to the lower amount of Zr^4+^ sites available to catalyze the reaction and to the narrower pore system, which introduce severe diffusion limitations to reactants and products and reduce the catalytic performance.

### 2.2. Reduction of 2-methylcyclohexanone (2MeCH = O)

The presence of a methyl group in position 2 in 2MeCH = O introduces a severe steric hindrance to the reduction of the carbonyl group, so a lower catalytic activity can be expected with respect to 3MeCH = O, as shown in Table 2.

As expected, the conversion of 2MeCH = O was very small over UiO-66, even at 120 °C (13% conversion after 24 h using 2-BuOH as solvent). In line with UiO-66, 2MeCH = O conversion was also much lower than 3MeCH = O for Zr/SBA, Zr-β and Ti-β (compare results in Table 1 and Table 2): 3.6% vs 94.1% conversion after 6 h for Zr/SBA; 6.1% vs 54.4% conversion after 0.5 h over Zr-β; and 8.8% vs 25.8% conversion after 6 h for Ti-β. Despite the lower activity of all the above catalysts, the reaction diastereoselectivity is still governed by a similar transition state selectivity as that seen in 3MeC = O reduction, as shown in Figure 3. That is, formation of the less bulky transition state is favored inside the small pores of Zr-β and Ti-β zeolites, with hydride transfer in the equatorial position and yielding the less thermodynamically stable *cis*-2MeCH-OH product; while the opposite situation is found in Zr/SBA. Nevertheless, the diastereoselectivities attained were around 55–60%; i.e., considerably lower than in the case of 3MeCH = O reduction.

Concerning MOF-808, the catalytic activity observed for the reduction of 2MeCH = O was practically the same than for 3MeCH = O when ^i^PrOH was used as a hydride source: full ketone conversion was attained after 6 h of reaction in both cases, clearly surpassing the activity of other catalysts described so far. As in wide pore Zr/SBA, the main product obtained was the thermodynamic alcohol, *trans*-2MeCH-OH, though in this case, the selectivity was also low (53%). A certain decrease in activity was observed when ^i^PrOH was replaced by 2-BuOH as a hydride source, although full ketone conversion was still achieved after 24 h of reaction time. Interestingly, the diastereoselectivity was inversed, and in this case, the *cis*-alcohol was obtained with 61% selectivity. A similar inversion of selectivity depending on the size of the reducing alcohol has been observed before using AlTPPCl catalyst and iso-borneol or ^i^PrOH as the reducing agent. [31]

To the best of our knowledge, the catalytic activity for the MPV reduction of 2MeCH = O is considerably better than any other catalysts reported so far. Therefore, it seems that MOF-808 features an optimum interplay between the structure of the active site and cavity effects that allows converting strongly hindered alcohols (such as ketones with methyl groups in *ortho* position) very efficiently.

### 2.3. Reduction of 2-phenylcyclohexanone (2PhCH = O)

To explore further the potential of MOF-808, we have considered the introduction of still bulkier substituents in the *ortho* position, a phenyl group. Table 3 summarizes the results obtained for the 2PhCH = O reduction over various catalysts using ^i^PrOH and 2-BuOH as hydride sources.

According to the above results, MOF-808 proved to be an outstanding catalyst for the MPV reduction of 2PhCH = O, despite the presence of the bulky phenyl ring in the *ortho* position to the ketone. With ^i^PrOH, full conversion was obtained after 24 h at 80 °C, with an excellent selectivity (90%) to *cis*-2phenylcyclohexan-1-ol (*cis*-2PhCH-OH). Very similar results were obtained when ^i^PrOH was replaced by 2Bu-OH (97% conversion after 24 h with 94% *cis* selectivity). Upon increasing the temperature to 120°C, the reaction rate sharply increased while the excellent *cis* selectivity was preserved. Compared to the reduction of 2MeCH = O, replacing the methyl group with a phenyl group in position 2 increases the steric hindrance. Therefore, the energy difference between TS-A (with the substituent pointing away from the MOF walls) and TS-B is larger for the bulkier substituent, resulting in a stronger preference for the *cis* product: 94% vs 61% *cis* selectivity for 2PhCH = O and 2MeCH = O, respectively (2-BuOH at 80 °C). Note also that, unlike in the reduction of 2MeCH = O, we did not observe an inversion of selectivity from *trans* to *cis* when the reducing alcohol was changed from ^i^PrOH to 2-BuOH (see Table 3). This is due again to the larger size of the phenyl group with respect to the methyl group, which prevents the formation of *trans*-PhCH-OH even with the smallest alcohol.

In a previous work, [23] Al(O^i^Pr)_3_ was reported to convert 2PhCH = O quantitatively after 19 h of reaction with 83% diastereoselectivity to *cis*-2PhCH-OH. In that work, the reaction was carried out at 50 °C in a 4:6 (*v/v*) mixture of ^i^PrOH/toluene. However, all our attempts to reproduce those results failed, observing virtually no conversion of 2PhCH = O under the above conditions. However, when the experiment was carried out under our reaction conditions (2-BuOH at 120 °C), Al(O^i^Pr)_3_ afforded 43% conversion of 2PhCH = O after 24 h of reaction, with a slightly lower *cis* diastereoselectivity of 74%. In any case, MOF-808 clearly outperformed Al(O^i^Pr)_3_ in terms of both catalytic activity and selectivity to the *cis* alcohol.

To put the catalytic results obtained with MOF-808 into perspective, we carried out the MPV reduction of 2PhCH = O using other catalysts for comparison. Thus, a very low conversion was attained with UiO-66, with a considerably lower *cis* selectivity (76%, see entry 5). Analogously, Zr-β zeolite also afforded a low conversion and similar *cis* selectivity (entry 6). In both cases, the low catalytic activity observed can be attributed to the large dimensions of 2PhCH = O with respect to the pore openings of the solid catalysts, thus preventing the reaction to take place inside the pores. Other zirconium catalysts, such as commercial ZrO_2_ nanopowder (Sigma-Aldrich < 100 nm) or Zr(O^i^Pr)_4_*^i^PrOH did not produce any measurable amount of 2PhCH-OH after 24 h of reaction (entries 7 and 8). Finally, when NaBH_4_ was used as a reducing agent (reaction conditions shown in Table 3 footnote), full reduction of 2PhCH = O was attained in 2 h, with a moderate selectivity to the *trans* alcohol; i.e., NaBH_4_ and MOF-808 afforded the opposite selectivity. In the case of carbonyl reduction with NaBH_4_, an axial attack of the hydride takes place from the less hindered face (see Figure 4). Meanwhile, in the case of the MPV reduction over MOF-808, the preference for the *cis* alcohol is again explained by the preferred formation of TS-A over TS-B, similar to what was observed for 2MeCH = O (see Figure 3).

### 2.4. Stability and Reusability of MOF-808

To complete the characterization of the catalytic activity of MOF-808, its recyclability was evaluated for the reduction of 3MeCH = O upon several consecutive reaction cycles. At the end of each catalytic cycle, the solid catalyst was recovered by filtration, washed with ethanol and dicloromethane, and dried at room temperature before use in the next run. As it can be seen in Appendix A, the catalytic activity of MOF-808 is preserved for at least 4 consecutive runs. Meanwhile, the diastereoselectivity measured at the end of the experiment was 70% (*cis*); that is, virtually unchanged with respect to that obtained in the first run. Moreover, the XRD of the solid recovered after 4 catalytic cycles showed that the material preserves the crystallinity under the reaction conditions used (see Appendix A).

## 3. Materials and Methods

All chemical reagents were purchased from Sigma-Aldrich and used without further purification.

### 3.1. Synthesis of Catalysts

**MOF-808**: Pristine zirconium MOF-808 material was prepared with slight modifications from an earlier reported procedure by Furukawa et al. [32]. Briefly, two solutions were prepared containing 242.5 mg of ZrOCl_2_*8H_2_0 (0.75 mmol), 105 mg of trimesic acid (0.5 mmol) and 22.5 mL of a DMF/HCO_2_H 1:1 (*v/v*) mixture. The solutions were mixed and transferred into a Teflon-lined autoclave and heated inside an oven at 130 °C for 48 h. After cooling down to room temperature, the material was recovered by centrifugation and washed for 3 days with DMF (changing the solvent 2 times per day) and for 3 days with EtOH (changing the solvent 2 times per day). After removing the solvent by centrifugation, the solid was dried in air.

**UiO-66**: The reference zirconium terephthalate UiO-66 material was prepared according to the reported procedure [33]. Briefly, 750 mg of ZrCl_4_ and 740 mg of terephthalic acid were dissolved in 90 mL of DMF (Zr:ligand:DMF molar ratio of 1:1:220) and the solution was kept in a closed round-bottom flask at 80 °C for 12 h without stirring, followed by another 24 h at 100 °C in an oil heating bath. The resulting material was recovered by filtration and washed thoroughly with fresh DMF. Then, the solid was washed three times by soaking in dichloromethane for 3 h. Finally, the solid was recovered by filtration and dried under vacuum. X-ray diffraction (PhillipsX’Pert, Cu Kα radiation) was used to confirm the expected structure type and high crystallinity of the material. The amount of missing linker defects in this UiO-66 sample was estimated to be ca. 7% from the corresponding TGA curve, following the method proposed by Valenzano et al. [34].

### 3.2. General Procedure for the MPV Reduction of Substituted Cyclohexanones

Specific reaction conditions for the MPV reduction of substituted cyclohexanones are indicated in the footnotes of each table. The general procedure used was as follows. Substituted cyclohexanone (0.1 mmol) was dissolved in 0.5 mL of alcohol (^i^PrOH or 2-BuOH). The solution was added to the catalyst (10 mg, 14 mol% with respect to cyclohexanone) under a nitrogen atmosphere inside a 2 mL glass batch reactor charged with a magnetic stirrer. The reactor was closed and heated in a metal heating block to the desired temperature while stirring. Aliquots were taken at specified time intervals and the course of the reaction was followed by GC analysis. In the case of 2MeCH = O and 2PhCH = O reduction, the analysis was carried out on an Agilent 7890A instrument equipped with an FID detector and a DB5 column (30 m × 0.25 mm × 0.25 µm). In the case of 3MeCH = O, a DB-WAX column was instead used (15 m × 0.32 mm × 0.25 µm).

## 4. Conclusions

In this work, the excellent performance of MOF-808 as a heterogeneous catalyst has been demonstrated for the MPV reduction of various substituted cyclohexanones, with a special emphasis on evidencing the diastereoselective properties of the catalytic process. Thus, the presence of a methyl group in position 3 does not hinder the MPV reaction, and the cyclohexanone is readily reduced over MOF-808 with either ^i^PrOH or 2-BuOH as a hydride source. The methyl group has a clear directing role towards the selective formation of the *cis* alcohol, in which both the -CH_3_ and -OH groups are located in the most thermodynamically favorable equatorial position. The opposite stereoselectivity reported was for narrow pore zeolites, and this was attributed to the restricted space available inside the pores, which precluded the formation of the bulkier transition state leading to the *cis* alcohol. In this sense, MOF-808 behaves similar to large mesoporous silicates.

Reduction of 2MeCH = O is a much more demanding reaction, due to the close proximity of the methyl group to the ketone which slows down the hydride transfer. The final selectivity to *cis*- or *trans*-alcohol is rather low and depends on the alcohol used: a bulkier alcohol (2-BuOH) favors the formation of the less hindered transition state, leading to *cis*-2MeCH-OH, whereas the other isomer is formed using a smaller alcohol (^i^PrOH).

A similar effect is observed for the reduction of 2PhCH = O, but the larger size of the phenyl group with respect to the methyl group introduces a higher steric control of the reaction, leading to significantly higher diastereoselectivities, up to 94% in favor of the *cis* alcohol. Interestingly, the catalytic activity of MOF-808 is by far higher than other solid catalysts, such as zeolites, ZrO_2_, or Al- and Zr-alkoxides.

## Data Availability

Not applicable.

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
