# Peer review of "MOF-808 as a Highly Active Catalyst for the Diastereoselective Reduction of Substituted Cyclohexanones"

_molecules, 2022, doi:10.3390/molecules27196315_

Round 1

Reviewer 1 Report

The authors reported reduction of substituted cyclohexanones catalyzed by MOF-808. The results are somewhat interesting, however, only the experimental results were listed. Further studies and discussion are required before consideration of publication. 

1. The most concern is the reduction process/mechanism. For example, there are no any evidences for the TS-A and TS-B given Schemes, at least the coordination of alcohol and substrate with Zr center must be proved. 

2.  An important point for heterogeneous catalytic reaction is the stability and reutilization of the catalyst. The catalyst after the catalytic reaction should be characterized and reused for another round of catalytic reaction. 

3. IR in Figure 1c is not acceptable. 

Author Response

Responses to Reviewer 1

  1. The most concern is the reduction process/mechanism. For example, there are no any evidences for the TS-A and TS-B given Schemes, at least the coordination of alcohol and substrate with Zr center must be proved. 

The structures shown in the Schemes for TS-A and TS-B are based on the generally accepted mechanism for the MPV reaction, through the formation of a six membered ring, which have been extensively supported by both experimental and theoretical means. In the particular case of MOFs, there is a vast literature on this topic, which demonstrates the interaction of both de alcohol and the carbonyl compound with the metal centers of the material: see for instance Molecular Catalysis 502 (2021) 111405; or ACS Catal. 2020, 10, 3720. We have clarified this point in the revised manuscript by adding the following sentence:

Thus, two transition states can be considered based on the most generally accepted mechanism for the MPV reduction of ketones over Lewis acid catalysts (Scheme 2) [20]” (lines 89-91 in p. 2)

  1. An important point for heterogeneous catalytic reaction is the stability and reutilization of the catalyst. The catalyst after the catalytic reaction should be characterized and reused for another round of catalytic reaction. 

We agree with the Referee that catalysts stability and reusability are of paramount important. Therefore, we have added a new section in the revised manuscript addressing this important point (section 2.4 in p. 7).

  1. IR in Figure 1c is not acceptable. 

This is the FTIR spectrum of a freshly prepared MOF-808 measured in ATR mode in a Bruker Tensor 27 instrument equipped with a single reflection diamond ATR accessory. The bands of this spectrum are totally in agreement with those reported before for MOF-808: absorption bands at 1620, 1525 and 1436 cm−1 due to the aromatic rings of the organic linker, and a strong Zr-O peak at 650 cm-1, indicating the coordination of Zr to the carboxyl groups of the linker. We sincerely don’t see what is the problem with this spectrum.

Reviewer 2 Report

The manuscript entitled „MOF-808 as a highly active catalyst for the diastereoselective reduction of substituted cyclohexanones” by Xamena et al. presents the catalytic application of Zr-containing MOF-808 in the Meerwein-Ponndorf-Verley reduction of substituted cyclohexanones. The report clearly demonstrates the successful utilization of the above catalyst in the hydrogenation process and its advantageous features over other types of solid catalysts. In the reduction of 2-Me-cyclohexanone unprecedentedly high activity could be obtained. In addition to that, mechanistic considerations are also presented to explain the different catalytic outcomes achieved by the different catalytic systems tested. The manuscript is generally well-written and the conclusions drawn seems to be correct. According to my opinion, the MS could be published in Molecules after addressing the following minor issues.

1.      page 1, line 38: “is” should be written instead of “in”

2.      p2, line 44: “and” instead of “an”?

3.      I believe that the abbreviations “MeCH=O” or “PhCH=O” are quite disturbing for the reader, as for the first look they could be acetaldehyde and benzaldehyde, respectively. Additionally, in page 2, cyclohexanones are abbreviated as CHs and not CH=Os. Please, correct!

4.      I think the authors used racemic starting materials, so the cis-alcohol in line 97 can not only be the (1S,3R) isomer but the (1R,3S) too.

5.      In tables conversion should be given in % instead of mol%. In the same way diastereoselectivity could be given in % only in the first row of the table.

6.      In the footnote of the tables the amount of the starting material and ratio of the catalyst relative to that, should be given.

7.      Several catalytic reactions are performed in 2-BuOH at 120 °C. According to the experimental section, the reactions are conducted “inside a 2 mL glass batch reactor charged with a magnetic stirrer” that is a closed vessel.  As the boiling point of 2-BuOH is ~99 °C, are the previously mentioned reactions performed under pressure? Please, clarifiy!

8.      p7, line 229: „measurable” instead of „sensible”?

9.      p7, line 242: „slight” instead of „slightly”

10.  p7, line 266: „14 mol% with respect to CH”, Please, give exactly the masses of the catalysts used.

11.  p7, line 268: „taken” instead of „takne”

12.  Please, give the exact GC conditions (eg. temperature program, gas pressures/fluxes etc.) for the analysis of the reaction mixtures and the retention times of the starting materials as well as the products under these determination conditions.

Author Response

Responses to Reviewer 2

The manuscript entitled „MOF-808 as a highly active catalyst for the diastereoselective reduction of substituted cyclohexanones” by Xamena et al. presents the catalytic application of Zr-containing MOF-808 in the Meerwein-Ponndorf-Verley reduction of substituted cyclohexanones. The report clearly demonstrates the successful utilization of the above catalyst in the hydrogenation process and its advantageous features over other types of solid catalysts. In the reduction of 2-Me-cyclohexanone unprecedentedly high activity could be obtained. In addition to that, mechanistic considerations are also presented to explain the different catalytic outcomes achieved by the different catalytic systems tested. The manuscript is generally well-written and the conclusions drawn seems to be correct. According to my opinion, the MS could be published in Molecules after addressing the following minor issues.

  1. page 1, line 38: “is” should be written instead of “in”

Corrected

  1. p2, line 44: “and” instead of “an”?

Corrected

  1. I believe that the abbreviations “MeCH=O” or “PhCH=O” are quite disturbing for the reader, as for the first look they could be acetaldehyde and benzaldehyde, respectively. Additionally, in page 2, cyclohexanones are abbreviated as CHs and not CH=Os. Please, correct!

We agree that the abbreviations used might be a bit confusing at first sight. The abbreviation CH stands for cyclohexanone. We added 2Me, 3Me or 2Ph to specify the type of substituent (methyl or phenyl) and the position. Finally, we then add =O or –OH to indicate whether the molecule contains a carbonyl or an alcohol group. Following the indications of the Referee, in the revised manuscript we have removed the abbreviation CH alone for cyclohexanone, to avoid confusion.

  1. I think the authors used racemic starting materials, so the cis-alcohol in line 97 can not only be the (1S,3R) isomer but the (1R,3S) too.

Referee 2 is correct. We have corrected the sentence accordingly.

  1. In tables conversion should be given in % instead of mol%. In the same way diastereoselectivity could be given in % only in the first row of the table.

Corrected. We thank Referee 2 for this remark.

  1. In the footnote of the tables the amount of the starting material and ratio of the catalyst relative to that, should be given.

To avoid having table footnotes unnecessarily too long, detailed reaction conditions are provided in the Materials and Methods section, point 3.2.

  1. Several catalytic reactions are performed in 2-BuOH at 120 °C. According to the experimental section, the reactions are conducted “inside a 2 mL glass batch reactor charged with a magnetic stirrer” that is a closed vessel. As the boiling point of 2-BuOH is ~99 °C, are the previously mentioned reactions performed under pressure? Please, clarifiy!

The Referee is correct: the reactions above the boiling point of the solvent are indeed performed under autogenous pressure. 

  1. p7, line 229: „measurable” instead of „sensible”?

Corrected

  1. p7, line 242: „slight” instead of „slightly”

Corrected

  1. p7, line 266: „14 mol% with respect to CH”, Please, give exactly the masses of the catalysts used.

Corrected. The amount of catalyst used (10 mg) is now specified in the text.

  1. p7, line 268: „taken” instead of „takne”

Corrected.

Reviewer 3 Report

This is the interesting work and may become suitable for publication in Molecules after addressing the following comments. Hence, I recommend this work for publication in Molecules after a revision.

Cyclohexanone and its derivatives should be abbreviated as C=O and 2MeC=O.

Typographical and English corrections have to be taken care.

Experimental conditions may be given as a footnote in each Table.

The following references have to be cited (ChemCatChem, 2020, 12, 4732-4753; Trends Chem, 2020, 2, 454-466)

Authors may explain the way by which % diastereoselectivity is determined. 

Did authors check the stability of MOF-808 through reusability? 

Author Response

Responses to Reviewer 3

This is the interesting work and may become suitable for publication in Molecules after addressing the following comments. Hence, I recommend this work for publication in Molecules after a revision.

Cyclohexanone and its derivatives should be abbreviated as C=O and 2MeC=O.

The abbreviations used might be a bit confusing at first sight. The abbreviation CH stands for cyclohexanone. We added 2Me, 3Me or 2Ph to specify the type of substituent (methyl or phenyl) and the position. Finally, we then add =O or –OH to indicate whether the molecule contains a carbonyl or an alcohol group.

Typographical and English corrections have to be taken care.

We have thoroughly revised the entire manuscript for typographical errors.

Experimental conditions may be given as a footnote in each Table.

We have included detailed experimental conditions in the Materials and Methods section (point 3.2) to avoid having table footnotes unnecessarily too long.

The following references have to be cited (ChemCatChem, 2020, 12, 4732-4753; Trends Chem, 2020, 2, 454-466)

These relevant references have been included in the revised manuscript.

Authors may explain the way by which % diastereoselectivity is determined. 

The amount of each diastereoisomer formed was determined from the corresponding 1H NMR spectra of the reaction crude. This is now indicated in the table footnotes.

Did authors check the stability of MOF-808 through reusability? 

We have included a new section in the revised manuscript and two additional figures as Supporting Information addressing this specific point.

Round 2

Reviewer 1 Report

Figure S1c is not acceptable: It is not absorbance for the IR spectrum and the values in the vertical ordinate are  no meaning.